# Tiled-ClickSeq for targeted sequencing of complete coronavirus genomes with simultaneous capture of RNA recombination and minority variants

**Elizabeth Jaworski[1,2], Rose M Langsjoen[1], Brooke Mitchell[3,4], Barbara Judy[5], Patrick Newman[5], Jessica A Plante[3,6,7], Kenneth S Plante[3,6,7], Aaron L Miller[5], Yiyang Zhou[1], Daniele Swetnam[1], Stephanea Sotcheff[1], Victoria Morris[1], Nehad Saada[3,4], Rafael RG Machado[3,4], Allan McConnell[3,4], Steven G Widen[1,8], Jill Thompson[8], Jianli Dong[5,7], Ping Ren[4], Rick B Pyles[5], Thomas G Ksiazek[3,6], Vineet D Menachery[3,4,7], Scott C Weaver[3,4,7], Andrew L Routh[1,7,9]\***

[1]Department of Biochemistry and Molecular Biology, The University of Texas Medical Branch, Galveston, United States; [2]ClickSeq Technologies LLC, Galveston, United States; [3]World Reference Center for Emerging Viruses and Arboviruses, University of Texas Medical Branch, Galveston, United States; [4]Department of Microbiology and Immunology, The University of Texas Medical Branch, Galveston, United States; [5]Department of Pediatrics, University of Texas Medical Branch, Galveston, United States; [6]Department of Pathology, University of Texas Medical Branch, Galveston, United States; [7]Institute for Human Infections and Immunity, University of Texas Medical Branch, Galveston, United States; [8]Next-Generation Sequencing Core, The University of Texas Medical Branch, Galveston, United States; [9]Sealy Centre for Structural Biology and Molecular Biophysics, University of Texas Medical Branch, Galveston, United States

**\*For correspondence:**
alrouth@utmb.edu

**Abstract** High-throughput genomics of SARS-CoV-2 is essential to characterize virus evolution and to identify adaptations that affect pathogenicity or transmission. While single-nucleotide variations (SNVs) are commonly considered as driving virus adaption, RNA recombination events that delete or insert nucleic acid sequences are also critical. Whole genome targeting sequencing of SARS-CoV-2 is typically achieved using pairs of primers to generate cDNA amplicons suitable for next-generation sequencing (NGS). However, paired-primer approaches impose constraints on where primers can be designed, how many amplicons are synthesized and requires multiple PCR reactions with non-overlapping primer pools. This imparts sensitivity to underlying SNVs and fails to resolve RNA recombination junctions that are not flanked by primer pairs. To address these limitations, we have designed an approach called *'Tiled-ClickSeq'*, which uses hundreds of tiled-primers spaced evenly along the virus genome in a single reverse-transcription reaction. The other end of the cDNA amplicon is generated by azido-nucleotides that stochastically terminate cDNA synthesis, removing the need for a paired-primer. A sequencing adaptor containing a Unique Molecular Identifier (UMI) is appended to the cDNA fragment using click-chemistry and a PCR reaction generates a final NGS library. Tiled-ClickSeq provides complete genome coverage, including the 5'UTR, at high depth and specificity to the virus on both Illumina and Nanopore NGS platforms. Here, we analyze multiple SARS-CoV-2 isolates and clinical samples to simultaneously characterize minority variants, sub-genomic mRNAs (sgmRNAs), structural variants (SVs) and D-RNAs. Tiled-ClickSeq therefore

provides a convenient and robust platform for SARS-CoV-2 genomics that captures the full range of RNA species in a single, simple assay.

## Introduction

Virus genomics and next-generation sequencing (NGS) are essential components of viral outbreak responses (*Grubaugh et al., 2019b*). Reconstruction of consensus genetic sequences is essential to identify adaptations correlated with changes in pathogenicity or transmission (*Gussow et al., 2020*). In addition to single nucleotide variations, studies of SARS-CoV-2 have identified numerous genomic structural variants (SVs) (*Yi, 2020*) that arise due to non-homologous RNA recombination. SVs typically comprise small insertions/deletions that nonetheless allow the variant genome to independently replicate and transmit. Numerous SVs have been described for CoVs including deletions of the accessory open-reading frames (aORFs) (*Yc et al., 2020*; *Muth et al., 2018*) and changes in spike protein observed in the B.1.1.7 (Alpha) and other variants of concern (*Kemp et al., 2020*). Adaptation of SARS-CoV-2 also occurs during passaging in cell-culture, such as small deletions that arise near the furin cleavage site of spike protein during amplification on Vero cells (*Ogando et al., 2020*). These deletions can alter the fitness and virulence of SARS-CoV-2 isolates and thus must be genetically characterized at the within-culture population level prior to passaged stock use in subsequent studies.

Similar to SVs, non-homologous RNA recombination also gives rise to defective-RNAs (D-RNAs), also known as defective viral denomes (DVGs). D-RNAs have been observed in multiple studies of coronaviruses (CoVs), including mouse hepatitis virus (MHV) (*Makino et al., 1984*; *Makino et al., 1985*; *Makino et al., 1988a*; *Makino et al., 1988b*), bovine CoV (*Chang et al., 1994*), avian infectious bronchitis virus (IBV) (*Penzes et al., 1995*), human CoV 299E (*Viehweger et al., 2019*; *Banerjee et al., 2001*; *Joo et al., 1996*; *Kim et al., 1993*). We recently demonstrated that SARS-CoV-2 is >10 -fold more recombinogenic in cell culture than other CoVs such as MERS (*Gribble et al., 2021*) and generates abundant D-RNAs containing RNA recombination junctions that most commonly flank U-rich RNA sequences. D-RNAs may change the fitness, disease outcomes, and vaccine effectiveness for SARS-CoV-2 similar to other respiratory pathogens such as influenza and RSV (*Vignuzzi and Lopez, 2019*). Together, these findings highlight the need to identify these RNA species and their impact on SARS-CoV-2 infection and pathogenesis.

Whole genome sequencing can be achieved through a range of approaches including non-targeted (random) NGS of virus isolates amplified in cell culture or directly from patient samples. However, when input material is limited, low viral genome copy numbers necessitate a template-targeted approach followed by molecular amplification by PCR or iso-thermal amplification to generate sufficient nucleic acid for sequencing. Generally, these require knowledge of the virus genome and the design of pairs of primers that anneal to the target genome. Perhaps the most popular method for SARS-CoV-2 sequencing is the 'ARTIC' approach (*Tyson et al., 2020*), which can reliably identify SNVs and minority variants present in as little as 3 % of genomes (*Grubaugh et al., 2019a*). However, the requirement for pairs of primers constrains where amplicons can be designed and imparts sensitivity to single nucleotide variants (SNVs). Multiple PCR reactions containing different pools of paired-primers must also be performed in order to obtain cDNA amplicons of the correct size and to prevent the interaction or mis-priming of PCR primers. Importantly, pairs of primers that do not flank RNA recombination junctions will be unable to detect unexpected or unpredicted RNA recombinant species. Finally, paired-primer approaches also necessitate the re-design and validation of alternative sets of primer-pairs for each specific NGS platform used (e.g. Illumina amplicons are 200–500 nts, Nanopore amplicons are ~2000–5000 nts).

To address these limitations and optimize the ability of NGS to quantify all types of viral genetic variants, we have combined '*ClickSeq*' with tiled-amplicon approaches. ClickSeq (*Routh et al., 2015b*; *Jaworski and Routh, 2018*) is a click-chemistry-based platform for NGS that prevents artifactual sequence chimeras in the output data (*Gorzer et al., 2010*). Using ClickSeq, the 3'end of an amplified cDNA segment is generated by the stochastic incorporation of terminating 3' azidonucleotides (AzNTPs) during reverse transcription. A downstream adaptor is '*click-ligated*' onto the cDNA using copper-catalyzed azide-alkyne cycloaddition (CuAAC). Therefore, '*Tiled-ClickSeq*' only requires one template-specific primer per cDNA amplicon. To achieve whole genome sequencing of a virus isolate or sample, multiple tiled primers are designed evenly along the virus genome. Only one

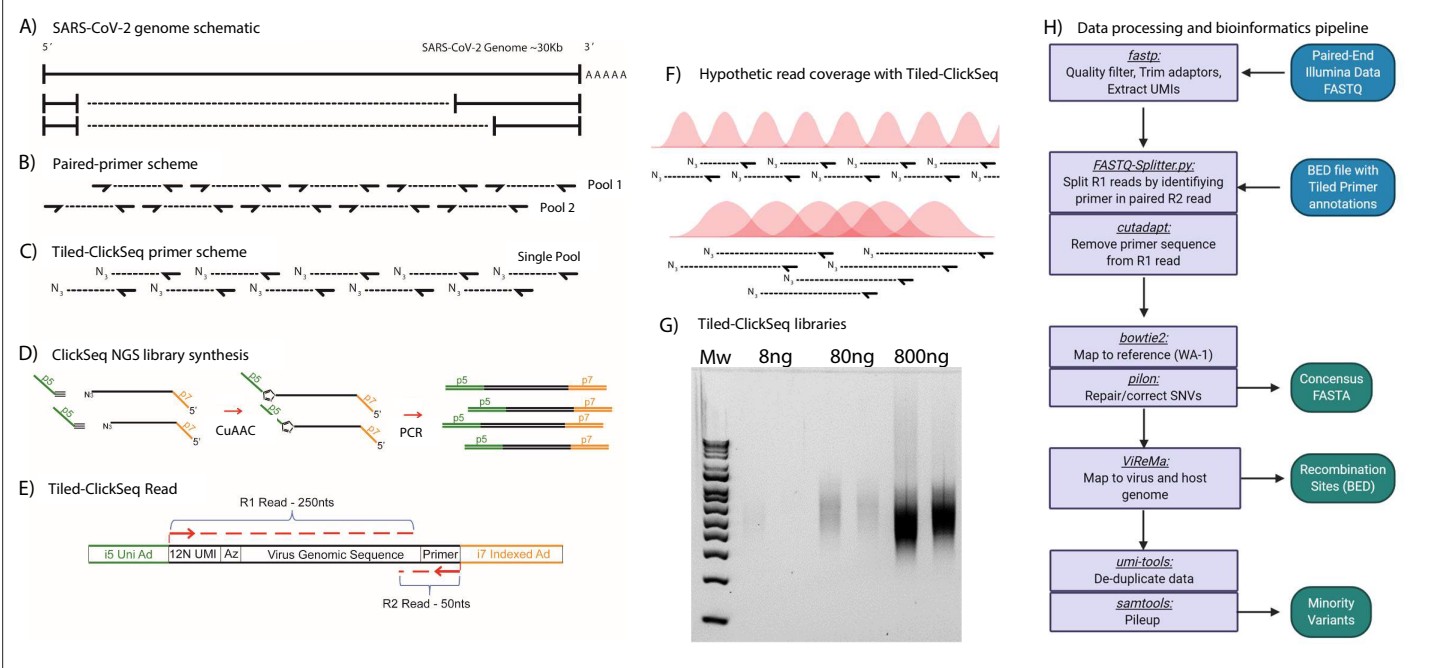

**Figure 1.** Schematic of Tiled-ClickSeq and computational pipeline: (**A**) Schematic of SARS-CoV-2 genome with two examples of sub-genomic mRNAs. (**B**) Paired-primer approaches typically generate short amplicons flanked by upstream and downstream primers that are PCR amplified in non-overlapping pools. (**C**) Tiled-ClickSeq uses a single pool of primers at the reverse-transcription step with the upstream site generated by stochastic termination by azido-nucleotides. (**D**) 3'-Azido-blocked single-stranded cDNA fragments are 'click-ligated' using copper-catalyzed azide alkyne cycloaddition (CuAAC) to hexynyl functionalized Illumina i5 sequencing adaptors. Triazole-linked ssDNA is PCR amplified to generate a final cDNA library. (**E**) The structure of the final cDNA is illustrated indicating the presence of the i5 and i7 adaptors, the 12 N unique molecular identifier (UMI), the expected location of the triazole linkage, and the origins of the cDNA in the reads including the tiled primer-derived DNA, which is captured using paired-end sequencing. (**F**) The hypothetical read coverage over a viral genome is indicated in red, yielding overlapping 'saw-tooth' patterns of sequencing coverage. Longer fragment lengths with more extensive overlapping can be obtained using decreased AzNTP:dNTP ratios. (**G**) Final cDNA libraries are analyzed and size-selected by gel electrophoresis (2 % agarose gel). Duplicates of libraries synthesized from 8, 80, and 800 ng of input SARS-CoV-2 RNA input are shown. (**H**) Flowchart of the data processing and bioinformatic pipeline. Input data is in Blue, output data are in Green, scripts/processes are Purple.

pool of RT-primers is required, even when > 300 template-specific primers and their corresponding cDNA amplicons are generated in the same reaction. This simplifies the assay design, and importantly removes constraints imposed in paired-primer strategies (*Itokawa et al., 2020*). Furthermore, the same primer set can be used for both Illumina and Nanopore platforms even when requiring different cDNA amplicon sizes. The library construction allows for additional quality control features including the use of unique molecular identifiers (UMIs) in the 'click-adaptor' as well as the ability to identify each RT-primer that gives rise to specific cDNA amplicon when using paired-read NGS.

Here, we utilize the Tiled-ClickSeq method to analyze multiple isolates of SARS-CoV-2 both from cell-culture and clinical specimens used in routine diagnostics for COVID19 and demonstrate that '*Tiled-ClickSeq*' accurately reconstructs full-length viral genomes. The method also captures recombinant RNA species including sgmRNAs, SVs, and D-RNAs. Overall, Tiled-ClickSeq therefore provides a convenient and robust platform for full genetic characterization of viral isolates.

## Results
### Overview of sequencing strategy

Most tiled approaches for complete viral genomes sequencing from viral isolates require the design of pairs of primers that generate pre-defined overlapping amplicons in multiple pools (*Figure 1A and B*). However, this can prevent the detection of recombinant viral genomic materials such as sub-genomic mRNAs (sgmRNAs) or Defective-RNAs (D-RNAs). To overcome these challenges, we designed a template directed tiled-primer approach to reverse transcribe segments of the SARS-CoV-2 genome

based upon the 'ClickSeq' method for NGS library synthesis (*Routh et al., 2015b*). Instead of random-hexamer or oligo-dT primers as used in ClickSeq and Poly(A)-ClickSeq, respectively (*Routh et al., 2017*), we use multiple 'tiled' RT-primers designed at regular intervals along the viral genome (*Figure 1C*). In '*Tiled-ClickSeq*', pooled primers initiate a reverse transcription in a reaction that has been supplemented with 3'-azido-nucleotides (AzNTPs). This yields stochastically terminated 3'-azido-cDNA fragments, which can be click-ligated onto a hexynyl-functionalized Illumina i5 sequencing adaptor (*Figure 1D*). After click-ligation, the single-stranded triazole-linked cDNA is PCR-amplified using indexing p7 adaptors to fill in the ends of the NGS library, yielding the final library schema shown in *Figure 1E*. We designed the click-adaptor with an additional 12 random nucleotides at its 5' end. As each adaptor can only be ligated once onto each unique cDNA molecule, this provides a unique molecular identifier (UMI) (*Jabara et al., 2011*). Due to the stochastic termination of cDNA synthesis in the RT step, a random distribution of cDNA fragments is generated from each primer, giving rise to the hypothetical read coverage depicted in *Figure 1F*. The lengths of these fragments, and thus the obtained read coverage can be optimized to ensure overlapping read data from each amplicon by adjusting the ratio of AzNTPs to dNTPs in the RT reaction (*Routh et al., 2015b*). With this approach, we found that we could robustly make NGS libraries from as little as 8 ng of total cellular RNA with only 18 PCR cycles (*Figure 1G*). Final libraries are excised from agarose gels (300-600nt cDNA size), pooled, and are compatible with Illumina sequencing platforms. A computational pipeline was compiled into a batch script (*Source data 2*) depicted by the flow-chart in *Figure 1H*.

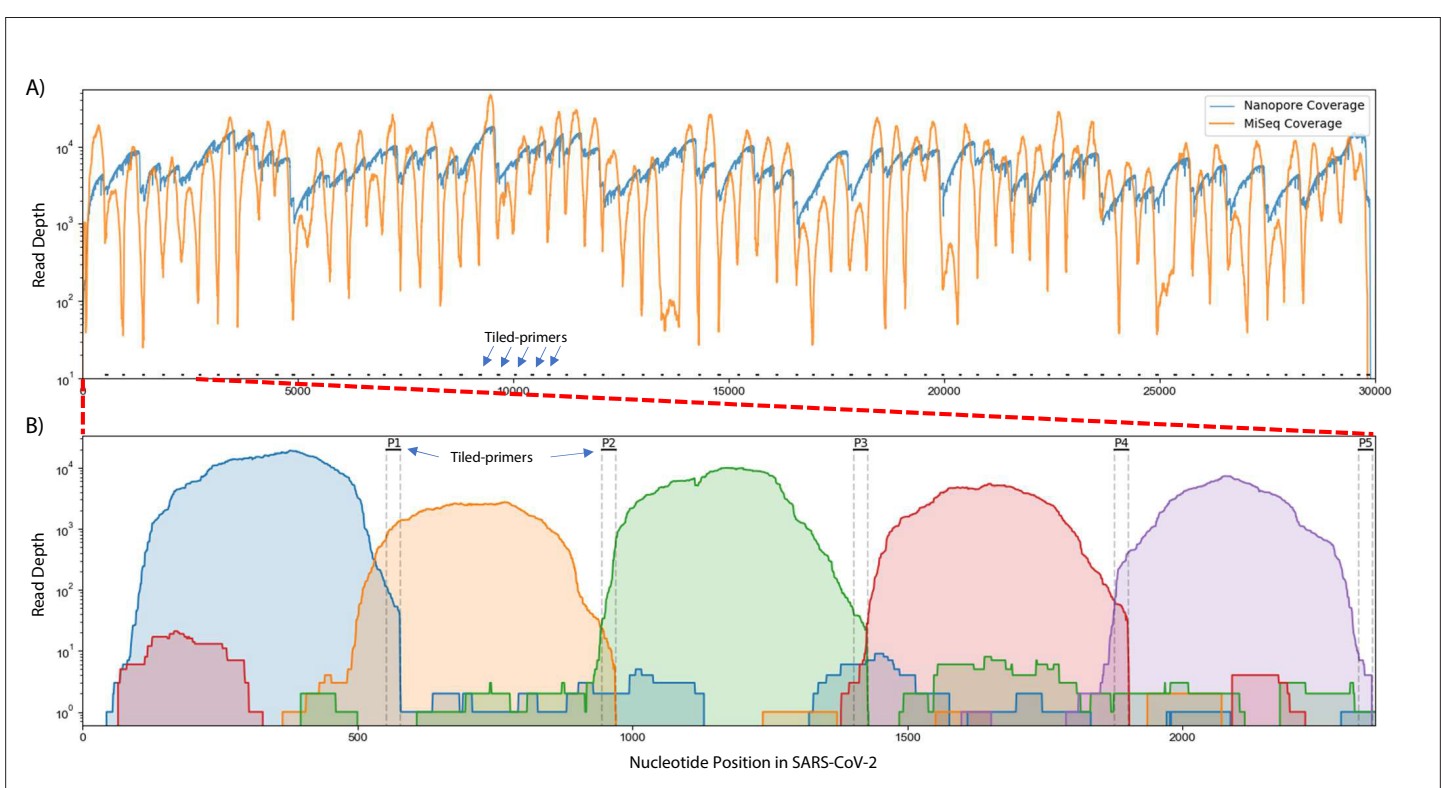

**Figure 2.** Read coverage over the SARS-CoV-2 genome using Tiled-ClickSeq. (**A**) Read coverage obtained from Tiled-ClickSeq over the whole viral genome is depicted when sequencing using an Illumina MiSeq (orange) or on an Oxford Nanopore Technologies MinION device (blue). A 'saw-tooth' pattern of coverage is observed with 'teeth' upstream of tiled-primers, indicated at the bottom of the plot by short black lines. (**B**) Zoomed in read coverage of nts 1–2400 of the SARS-CoV-2 genome with coverage of Illumina MiSeq reads from five individual primers coloured to illustrate coverage from downstream amplicons overlapping the primer-binding sites of upstream tiled-primers (Blue: Read coverage from primer 1; Orange: coverage from primer 2; Green: coverage from primer 3; Red: coverage from primer 4; Purple: coverage from primer 5).

The online version of this article includes the following figure supplement(s) for figure 2:

**Figure supplement 1.** Read coverage of tiled nanopore data over 12 SARS-CoV-2 isolates.

**Figure supplement 2.** Read coverage of tiled ARTIC data over 12 SARS-CoV-2 isolates: (**A**) Read coverage obtained from ARTIC sequencing protocol over the whole viral genome for 12 WRCEVA isolates is depicted when sequenced on an Illumina NextSeq.

## Validation with WA-1 strain

To test this approach, we obtained 200 ng RNA from an SARS-CoV-2 isolate deposited at the World Reference Center for Emerging Viruses and Arboviruses (WRCEVA) at UTMB (*Harcourt et al., 2020b*) and performed Tiled-ClickSeq using CoV2 primer pool v1 and a 1:35 AzNTP:dNTP mix. NGS libraries were sequenced on an Illumina MiSeq (2 × 150 reads). Reads were quality processed using *fastp* (*Chen et al., 2018*) and mapped to the virus genome using *bowtie2* (*Langmead and Salzberg, 2012*). A 'saw-tooth' pattern of read coverage over the genome was generated (*Figure 2A*, orange plot) with 'teeth' appearing as expected upstream of each tiled primer. Peaks of coverage for each 'tooth' ranged from ~13,000 x to ~100 x. Overall, we obtained genome coverage >25 X from nucleotide 3–29823 (50nts from the 3' end of the genome). This depth is sufficient to reconstruct a consensus genome sequence that was found to be identical to that already deposited (MT020881) for this isolate (*Harcourt et al., 2020a*).

When using paired-end sequencing, the 'forward'/'R1' read is derived from the click-adaptor and contains the UMI. The 'reverse'/'R2' read is derived directly from the tiled primer (see schematic in *Figure 1E*). We wrote a custom python3 script to split all the forward 'R1' reads into multiple individual FASTQ files based upon which primer generated each fragment. The mapping coverage obtained from five individual tiled-primers is shown in *Figure 2B*. The coverage for each primer (denoted by individual colours in *Figure 2B*) spans approximately 500–600 nts and extends 5'-wards from the tiled RT-primer. Read coverage from each primer overlaps the read coverage of the upstream primer. This allows for continuous gap-free read coverage over the viral genome which, importantly, allows a downstream cDNA amplicon to provide sequence information over and beyond an upstream primer. Additionally, we can determine the frequency with which each primer either successfully maps to the viral genome, mis-primes from the host RNA, or gives rise to adaptor-dimers or other sequencing artifacts. This information can be used to identify primers that yield poor viral priming efficiency and therefore a more specific primer can be designed and substituted as needed.

For nanopore sequencing, we also synthesized Tiled-ClickSeq libraries but using a 1:100 AzNT-P:dNTP ratio to generate cDNA amplicons of increased lengths. We retained cDNA fragments > 600 nts, yielding a few nanograms of dsDNA. This library, though containing the Illumina adaptors, can nonetheless be used as input in the default Oxford Nanopore Technologies (ONT) Ligation-Sequencing protocol (LSK-109) that appends ONT adaptors directly onto the ends of A-tailed dsDNA fragments. We sequenced this library using an ONT MinION device and obtained 279,192 reads greater than 1kbp in length. These were mapped to the WA-1 viral genome using *minimap2* yielding continuous genome coverage (*Figure 2A*, blue). A similar profile of read coverage to the Illumina data was observed, with peaks of coverage upstream of tiled-primer sites. The deeper dips in coverage

**Table 1.** Read counts and mapping rates for random-primed versus Tiled-ClickSeq approaches.

| Sample | CT | ClickSeq reads | Virus mapped | % Viral Reads | Tiled v1 reads | Virus mapped | % Viral Reads |
|---|---|---|---|---|---|---|---|
| WRCEVA_00501 | 12.9 | 4,665,869 | 116,036 | 2.5% | 2,359,795 | 2,204,750 | 93.4% |
| WRCEVA_00502 | 12.9 | 4,989,513 | 118,260 | 2.4% | 1,962,581 | 1,820,925 | 92.8% |
| WRCEVA_00505 | 12.7 | 3,894,325 | 71,809 | 1.8% | 2,779,672 | 2,482,854 | 89.3% |
| WRCEVA_00506 | 12.5 | 4,979,989 | 108,532 | 2.2% | 2,395,750 | 2,148,256 | 89.7% |
| WRCEVA_00507 | 12.9 | 5,659,073 | 161,059 | 2.8% | 2,056,670 | 1,867,012 | 90.8% |
| WRCEVA_00508 | 16.8 | 3,987,009 | 91,452 | 2.3% | 1,787,418 | 1,433,005 | 80.2% |
| WRCEVA_00509 | 17.1 | 4,057,928 | 57,424 | 1.4% | 2,202,661 | 1,856,633 | 84.3% |
| WRCEVA_00510 | 16.2 | 5,328,829 | 65,281 | 1.2% | 2,040,332 | 1,601,544 | 78.5% |
| WRCEVA_00513 | 16.0 | 4,391,175 | 69,169 | 1.6% | 1,641,213 | 1,455,991 | 88.7% |
| WRCEVA_00514 | 12.9 | 4,340,084 | 84,211 | 1.9% | 2,089,241 | 1,902,748 | 91.1% |
| WRCEVA_00515 | 15.7 | 5,416853 | 102,179 | 1.9% | 2,205,166 | 1,915,129 | 86.8% |
| WRCEVA_00516 | 17.4 | 4,290,929 | 61,017 | 1.4% | 1,988,939 | 1,715,448 | 86.2% |

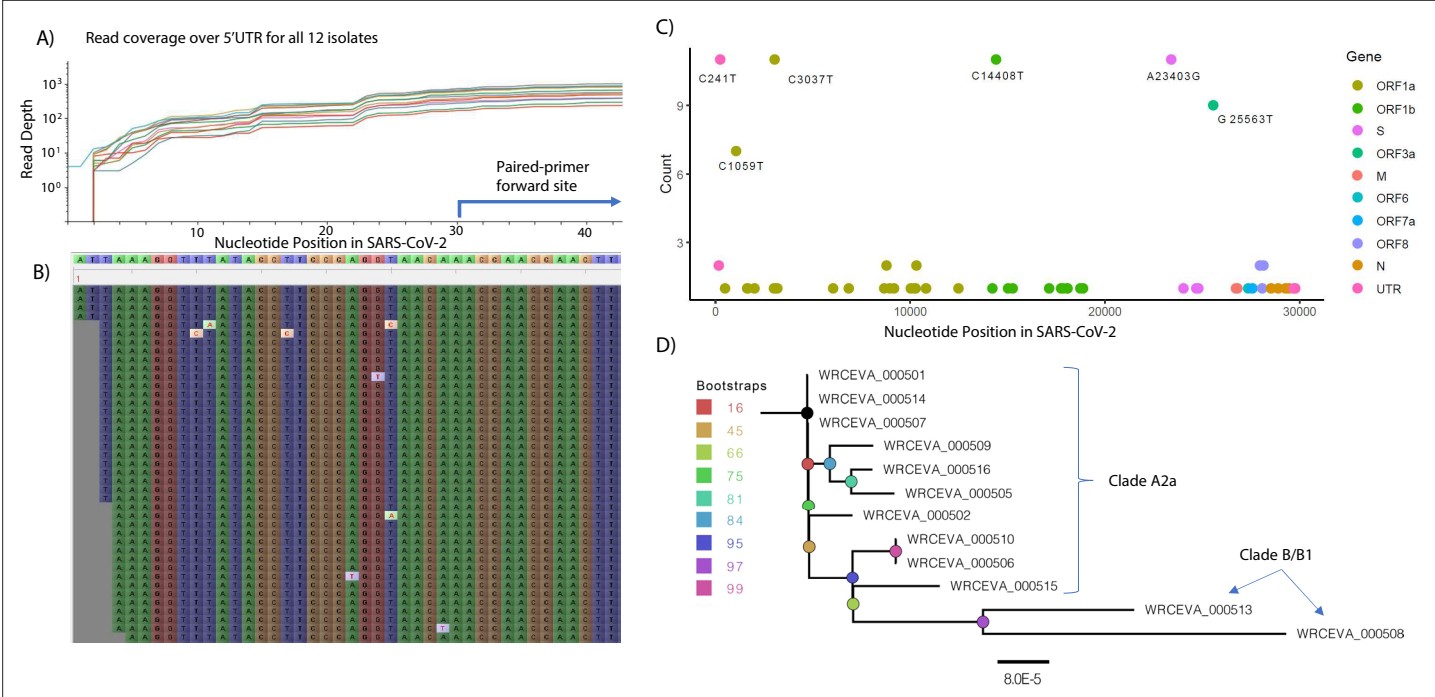

**Figure 3.** Genome Reconstruction of 12 SARS-CoV-2 isolates deposited at the World Reference Center for Emerging Viruses and Arboviruses (WRCEVA). (**A**) Read coverage is depicted over the 5' UTR of the SARS-CoV-2 genome for each isolate revealing capture of this region. The 5'-most primer from the ARTICv3 protocol at nts-30–54 is illustrated. (**B**) Snapshot of read data from Tiled-ClickSeq is depicted using the Tablet Sequencing Viewer from WRCEVA_000508 over the same region of the 5'UTR as A. (**C**) The most common single-nucleotide variants (SNVs) found in complete genome reconstructions from all 12 isolates are illustrated and colour-coded to depict the underlying viral protein. (**D**) Phylogenetic tree of 12 WRCEVA isolates with their corresponding clade indicated.

were avoided however, due to the longer reads lengths that give greater overlap between cDNA amplicons.

## Genome reconstruction of 12 isolates: ClickSeq, Tiled-ClickSeq, and Nanopore-Tiled-ClickSeq

To validate the suitability of Tiled-ClickSeq for whole virus genome reconstruction, we obtained RNA extracted from 12 outgrowth samples of SARS-CoV-2 deposited at WRCEVA from nasopharyngeal swabs collected between March and April 2020. We synthesized 12 Tiled-ClickSeq libraries and 12 random-primed ClickSeq libraries in parallel. These were submitted for sequencing on a NextSeq (2 × 150) yielding ~2–5 M reads per sample (*Table 1*). Random-primed ClickSeq data were quality-filtered and adaptor trimmed using *fastp* (*Chen et al., 2018Chen et al., 2018*) retaining only the forward R1 reads. Tiled-ClickSeq read data were processed and mapped following the scheme in *Figure 1H*.

In the Tiled-ClickSeq data, after UMI deduplication, each isolate had an average coverage between 4500 and 7500 reads and a coverage of 25 reads in greater than 99.5 % (29753/29903 nts) of the SARS-CoV-2 genome. Read coverage was also obtained covering the 5'UTR of each strain ( > 25 reads for all isolates from nucleotide three onwards (*Figure 3A and B*)). When using paired-primer approaches, the 5'UTR is ordinarily obscured by the 5'-most primer used in each pool (nts 30–54 for the ARTIC primer set depicted in *Figure 3A*). As the 5' end is resolved here due to stochastic incorporation of a single AzNTP in a template-specific manner, the entirety of the viral genome can be resolved. We reconstructed reference genomes from mapped reads using *pilon* (*Walker et al., 2014Walker et al., 2014*) requiring 25 x coverage for variant calling. In all cases, the reconstructed reference genomes were identical with or without controlling for PCR duplicates using the UMIs. We found 5–12 SNVs per viral genome (*Source data 3*), including the prevalent D614G (A23403G) spike adaptation, which enhances SARS-CoV-2 transmission (*Plante et al., 2020*), in 11 out of the 12 isolates (*Figure 3C*).

Genome reconstruction was similarly performed using the random-primed ClickSeq data reads. Identical genomes to the Tiled data were obtained for 11 out of 12 isolates, with only one SNV

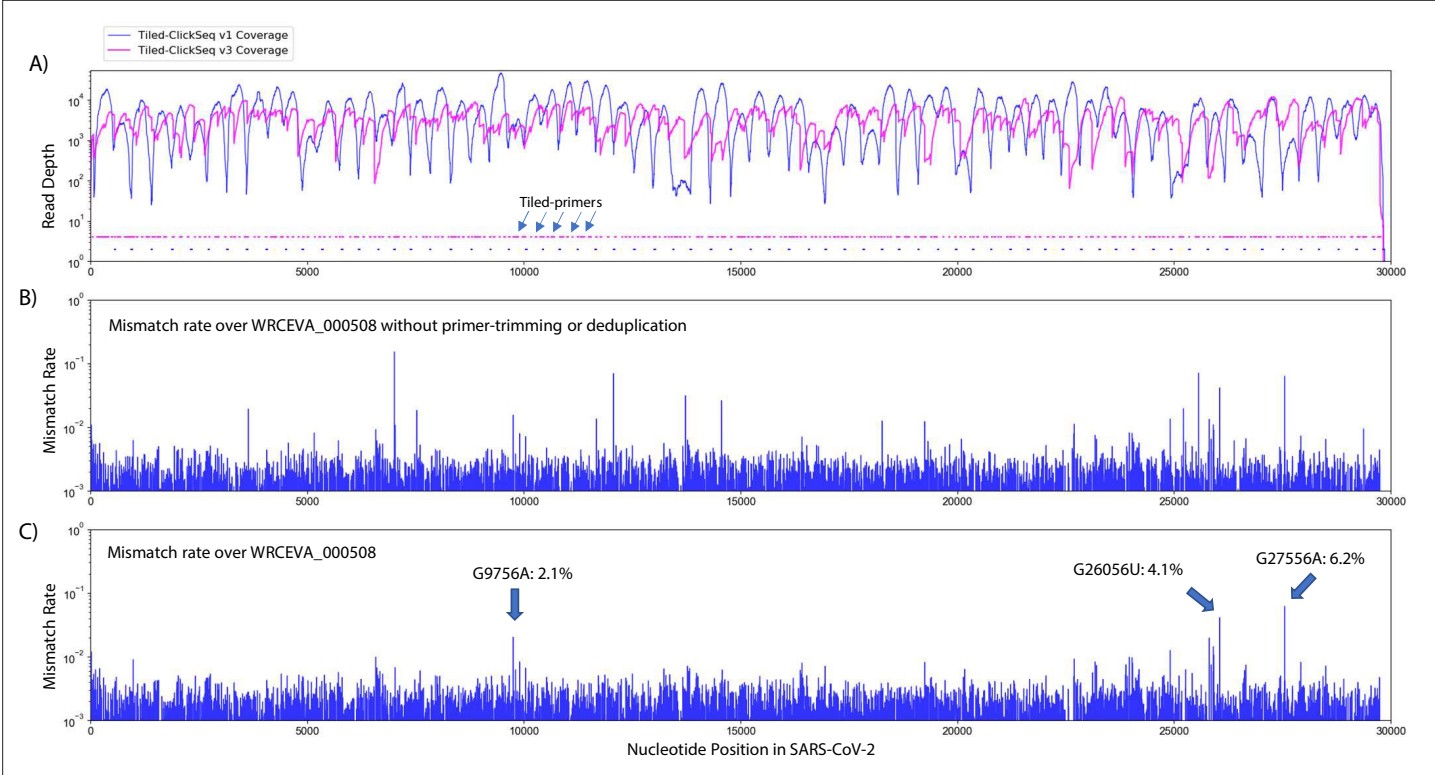

**Figure 4.** Additional tiled-primers improves read coverage and allows identification of minority variants. (**A**) Read coverage obtained from Tiled-ClickSeq over the whole viral genome is depicted using an Illumina MiSeq when using the original primers as in *Figure 2* (v1 - blue) or with an additional 326 tiled-primers (v3 - pink). Tiled-primers are indicated at the bottom of the plot by short blue (**v1**) or pink (**v3**) lines. (**B**) The rates of mismatching nucleotides found in mapped NGS reads is depicted across the SARS-CoV-2 genome for isolate WRECVA_000508 prior to trimming the tiled primers from forward/'R1' reads and without PCR deduplication. (**C**) The rates of mismatching is also depicted after data quality processing to remove PCR duplicates and primer-derived nucleotides in the reads, revealing three minority variants in this sample with frequencies > 2%.

The online version of this article includes the following figure supplement(s) for figure 4:

**Source data 1.** The frequency of all mapped nucleotides at each genome coordinate for each WRCEVA isolate is provided.

difference in one sample (WRCEVA _000510: T168C). In this case, the read coverage was too low in the random-primed data for *pilon* to report an SNV. Nevertheless, visual inspection of the mapped data revealed that all nucleotides at this locus were indeed C's, as reported for the Tiled-ClickSeq data. Phylogenetic tree reconstruction using NextStrain (*Hadfield et al., 2018*) placed 10 of the isolates in the A2a clade (*Figure 3D*) and two isolates (WRCEVA_00508, WRCEVA_00513) were Clade B/B1.

We also retained cDNA fragments > 600 bps from the Tiled-ClickSeq libraries and sequenced these using an ONT MinION device. We used the ONT native barcoding kit to multiplex the 12 samples and the Ligation-Sequencing protocol (LSK-109) to generate final libraries. Reads were mapped with *minimap2* (*Li, 2016*) yielding at least 100 x coverage over >99.6% of the genome for each isolate (*Figure 2—figure supplement 1*). Again, reference genomes were reconstructed from the mapped data using *pilon* (*Source data 3*). With the exception of WRCEVA_000514 which contained a single additional SNV (C14220T), the reference genomes reconstructed from the nanopore data were identical to those generated from the Tiled-ClickSeq Illumina data. These data illustrate that Tiled-ClickSeq performs as well as random-primed methods either on Illumina or Nanopore platforms for whole genome reconstruction.

To further validate our approaches, we used the well described ARTIC v3 protocol for amplicon sequencing of whole SARS-CoV-2 viral genomes (*Tyson et al., 2020*) using the same input RNA as above for Tiled-ClickSeq NGS library synthesis. In every case, the reported SNVs were identical between the ARTIC data and the Tiled-ClickSeq data. Read coverage over the viral genomes is illustrated in *Figure 2—figure supplement 2*.

## Minority variants

Our initial primer design (v1) (*Figure 4A*, blue plots) successfully yielded coverage suitable for complete genome reconstruction. However, some regions still received low coverage with fewer than a 100 deduplicated reads, preventing identification of minority variants in these regions. Therefore, we redesigned our primer scheme by adding an additional 326 primers (v2) previously reported (*Guo et al., 2020*) for tiled coronavirus sequencing (*Source data 1*) to make a pool comprising a total of 396 unique primers (v3). We re-sequenced the 12 WRCEVA isolates analyzed as described above plus an additional four that subsequently became available. An example of mapping coverage for isolate WCREVA_000508 is illustrated in *Figure 4A*, where the coverage over the viral genome is more even with less extreme ranges of read depth.

Using the R2 read, we can determine which primer gives rise to each R1 read and trim primer-derived nucleotides from the R1 read. This is an important quality control as it prevents the assignment (or failure thereof) of SNVs and/or the mapping of recombination events due to primer mis-priming. If reads are mapped without trimming away the primer-derived nucleotides found in the R1 read (as depicted in *Figure 4B*), we see numerous high frequency (2–50%) minority variants. The majority of these apparent minority variants overlap primer-target sites and are likely artefactual. Furthermore, the same high-frequency events are often seen across multiple independent samples. To control for

**Table 2.** Minority variants and rates ( > 2%) found across 16 WRCEVA isolates.

| Sample | Nt | Nuc | Read Depth | A | U | G | C | Variant Rate | Location | Result |
|---|---|---|---|---|---|---|---|---|---|---|
| WRCEVA_000501 | 12,049 | C | 2,116 | 0 | 95 | 1 | 2020 | 4.5% | ORF1ab | N3928K |
| WRCEVA_000502 | 10,207 | C | 2,240 | 0 | 118 | 0 | 2,122 | 5.3% | - | - |
| WRCEVA_000502 | 16,050 | U | 3,853 | 0 | 3,322 | 0 | 531 | 13.8% | - | - |
| WRCEVA_000502 | 17,489 | A | 4,597 | 4,433 | 162 | 1 | 1 | 3.6% | ORF1ab | E5742V |
| WRCEVA_000502 | 21,526 | A | 8,749 | 6,508 | 0 | 2,240 | 1 | 25.6% | ORF1ab | I7088V |
| WRCEVA_000503 | 14,220 | C | 1,638 | 1 | 463 | 0 | 1,174 | 28.3% | - | - |
| WRCEVA_000504 | 1,556 | A | 2,828 | 2,499 | 0 | 328 | 1 | 11.6% | ORF1ab | I431V |
| WRCEVA_000504 | 27,925 | C | 2,857 | 0 | 134 | 0 | 2,723 | 4.7% | ORF8 | T11I |
| WRCEVA_000507 | 19,515 | A | 2,393 | 2,295 | 1 | 97 | 0 | 4.1% | - | - |
| WRCEVA_000508 | 9,756 | G | 1,376 | 28 | 0 | 1,348 | 0 | 2.1% | ORF1ab | R3164H |
| WRCEVA_000508 | 26,056 | G | 2092 | 0 | 86 | 2006 | 0 | 4.1% | ORF3a | D222Y |
| WRCEVA_000508 | 27,556 | G | 2066 | 128 | 0 | 1938 | 0 | 6.2% | ORF7a | A55T |
| WRCEVA_000509 | 11,956 | C | 1962 | 0 | 199 | 0 | 1,763 | 10.1% | - | - |
| WRCEVA_000509 | 17,245 | C | 4,062 | 2 | 470 | 0 | 3,590 | 11.6% | ORF1ab | R5661C |
| WRCEVA_000509 | 18,005 | U | 5,408 | 1 | 4,949 | 458 | 0 | 8.5% | ORF1ab | L5915R |
| WRCEVA_000509 | 25,569 | U | 3,448 | 4 | 3,326 | 113 | 5 | 3.5% | - | - |
| WRCEVA_000509 | 27,919 | U | 839 | 0 | 809 | 0 | 30 | 3.6% | ORF8 | I9T |
| WRCEVA_000509 | 28,767 | C | 2011 | 0 | 109 | 0 | 1902 | 5.4% | N | T165I |
| WRCEVA_000511 | 3,003 | U | 2,880 | 79 | 2,787 | 1 | 13 | 2.7% | ORF1ab | V913E |
| WRCEVA_000511 | 10,738 | U | 4,580 | 0 | 4,440 | 0 | 140 | 3.1% | - | - |
| WRCEVA_000511 | 25,892 | U | 133 | 0 | 130 | 0 | 3 | 2.3% | ORF3a | I167T |
| WRCEVA_000511 | 28,001 | G | 1,414 | 1 | 29 | 1,384 | 0 | 2.1% | - | - |
| WRCEVA_000513 | 27,046 | C | 5,539 | 0 | 138 | 0 | 5,401 | 2.5% | M | T175M |
| WRCEVA_000514 | 11,603 | A | 5,405 | 5,075 | 0 | 330 | 0 | 6.1% | ORF1ab | M3780V |
| WRCEVA_000514 | 26,526 | G | 525 | 0 | 20 | 505 | 0 | 3.8% | M | A2S |

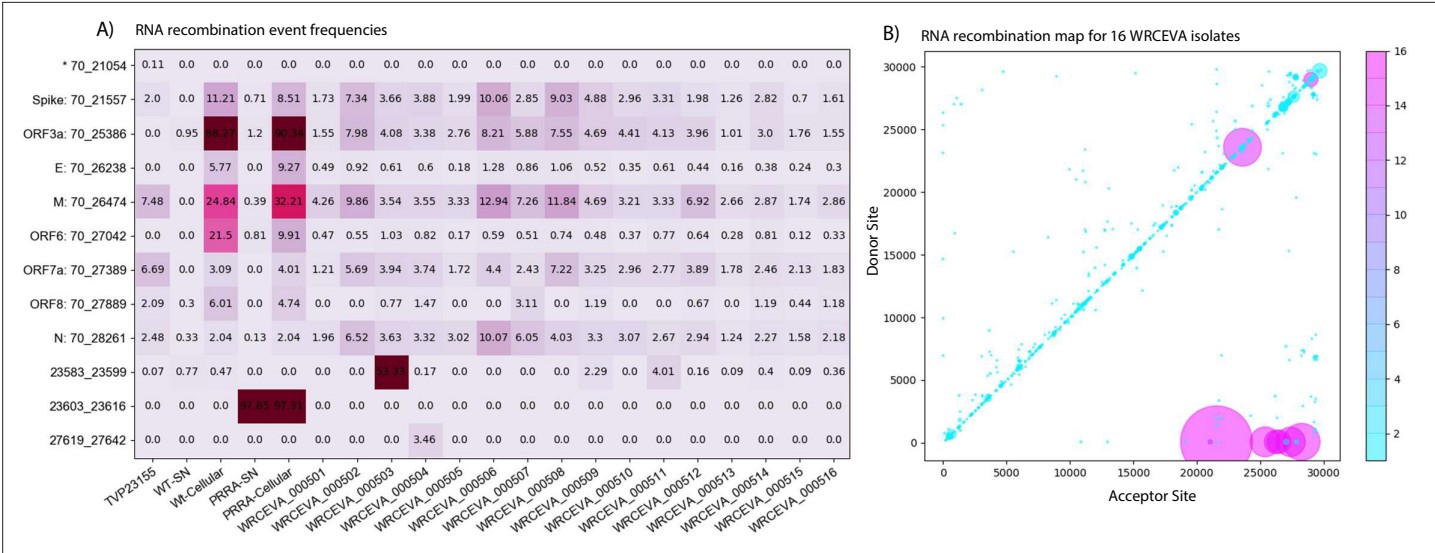

**Figure 5.** Tiled-ClickSeq identifies sub-genomic mRNAs, structural variants, and defective-RNAs. (**A**) A table of the most common RNA recombination events found using Tiled-ClickSeq to study 'World Reference Center for Emerging Viruses and Arboviruses' (WRCEVA) isolates. The recombination junctions are indicated on the left of the table, with their relative frequencies indicated in the table and colour-matched for each sample analyzed. All canonical sgmRNAs are found with their open-reading frame (ORF) indicated, in addition to one non-canonical sgmRNA (*). Three common structural variants including two deletions in spike protein and a deletion in ORF7a were also detected. (**B**) Unique RNA recombination events are plotted for 16 WRCEVA isolates as a scatter plots whereby the upstream 'donor' site is plotted on the y-axis and a downstream 'acceptor' site is plotted on x-axis. The read count for each unique RNA recombination event is indicated by the size of the point, while the number of samples in which this each RNA recombination event is found is indicated by the colour-bar. Insertions/duplication/back-splicing events are found above the x = y axis, while deletions and RNA recombination events yielding sgmRNAs are found below.

The online version of this article includes the following source data and figure supplement(s) for figure 5:

**Source data 1.** Snapshot of Tiled-ClickSeq reads from icSARS-CoV-2 delta-PRRA.

**Figure supplement 1.** Integrative Genomics Viewer (IGV) snapshot of Tiled-ClickSeq data over icSARS-CoV-2 delta PRRA: A full-view of the SARS-CoV-2 genome with mapped Tiled-ClickSeq reads is depicted using Integrative Genomics Viewer.

this, we mapped reads after trimming away primer-derived nucleotides from the R1 reads as per our pipeline described above (schematic in *Figure 1H*). Finally, to control for PCR duplication events, we make use of the UMIs embedded in the click-adaptor. The final de-duplicated mapped, primer-trimmed reads (*Figure 4C*) provide a robust readout of minority variants in these isolates (*Table 2* and *Figure 4—source data 1*). Across 10 WRCEVA isolates we found only 26 minority variants present at >2% all of which were unique within this dataset. Six isolates reported no minority variants at all.

## RNA recombination: sgmRNAs, structural variants, and defective RNAs

To characterize RNA recombination, we used our bespoke *ViReMa* pipeline (*Routh and Johnson, 2014*) to map RNA recombination events in NGS reads that correspond to either sgmRNAs, SVs, or D-RNAs. *ViReMa* can detect agnostically a range of expected and unusual RNA recombination events including deletions, insertions, duplications, inversions as well as virus-to-host chimeric events and provides BED files containing the junction sites and frequencies of RNA recombination events. We mapped the Tiled-ClickSeq data to the corrected reference genome for each WRCEVA isolate using *ViReMa*. We also took total cellular RNA and RNA extracted from the supernatants of *Vero* cells transfected with RNA derived from an in vitro infectious clone of SARS-CoV-2 (icSARS-CoV-2) (*Xie et al., 2020*). These clone-derived RNAs contained either the WT SARS-CoV-2, or were engineered with a deletion near the furin cleavage site of the spike protein, which we recently demonstrated is a common adaption to Vero cells and which alters SARS-CoV-2 pathogenesis in mammalian models of infection (*Johnson et al., 2021*).

The identities and frequencies of the 13 most abundant RNA recombination events are illustrated in *Figure 5A*. We found all the expected sgmRNAs previously annotated for SARS-CoV-2 (*Kim et al., 2020*) as well as non-canonical sgmRNAs. An overview of mapped data over the SARS-CoV-2

**Table 3.** Micro-indels and rates ( > 2%) found across 16 WRCEVA isolates.

| Sample | MicroInDel | Nucs | Variant Rate | Location | Result |
|---|---|---|---|---|---|
| WRCEVA_000502 | Δ519^523 | UGGUU | 2.2% | ORF1AB | Frameshift |
| WRCEVA_000504 | Δ29686^29,693 | CAGUGUGU | 3.5% | 3'UTR | - |
| WRCEVA_000505 | Δ519^523 | UGGUU | 2.9% | ORF1AB | Frameshift |
| WRCEVA_000506 | Δ519^523 | UGGUU | 3.8% | ORF1AB | Frameshift |
| WRCEVA_000509 | Δ1237^1,239 | UCA | 2.9% | ORF1AB | ΔH325 |
| WRCEVA_000510 | Δ686^694 | AAGUCAUUU | 5.1% | ORF1ab | ΔLSF141-143 |
| WRCEVA_000511 | Δ519^523 | UGGUU | 3.7% | ORF1AB | Frameshift |
| WRCEVA_000511 | Δ10811^10,813 | CUU | 3.1% | ORF1AB | ΔL3516 |
| WRCEVA_000512 | Δ29750^29,759 | GAUCGAGUG | 10.0% | 3'UTR | - |

illustrating large recombination events (depicted by the blue horizontal lines) is provided in *Figure 5— figure supplement 1*. We found that sgmRNAs were highly enriched in the cellular fractions from expressed icSARS-CoV-2 isolates (comprising >95% of the total viral genetic materials) but were relatively depleted in the supernatant fraction. This reflects a strong restriction of the packaging of these RNA species into virions. In the icSARS-CoV-2 samples, Tiled-ClickSeq and *ViReMa* accurately reported the expected deletion (Δ23603^23616). Interestingly, we also identified small structural variants (Δ23583^23599) in seven of the WRCEVA isolates with a frequency of 2–50%, similar to reports of the selection of variants containing deletions at this site after in vitro passaging on Vero cells (*Klimstra et al., 2020*). We also found a novel SV in one isolate (WRCEVA_000504: Δ27619^27642) present in 3.5 % of the reads resulting in an eight amino acid deletion in ORF7a. We additionally identified a small number of micro-indels (*Table 3*) in some isolates.

Finally, we observed thousands of RNA recombination events corresponding to D-RNAs (BED files for each sample are provided in *Figure 5—source data 1*). Despite their individual low frequencies, these events (displayed as a Recombination Heatmap in *Figure 5B*) reveals interesting features of D-RNAs of SARS-CoV-2. Apparent duplication events or insertions were most commonly observed with recombination events enriched around the 3'UTR of the genome, consistent with our previous characterization of RNA recombination in distinct coronavirus isolates including MHV, MERS and SARS-CoV-2 (*Gribble et al., 2021*). Finally, large deletions comprising RNA recombination events stretching from nucleotides ~ 6000–7000 to the 3'UTR were also observed, again, consistent with our previous observations. Altogether, these results demonstrate RNA recombination is a common and conserved feature of SARS-CoV-2 and that the emergence of D-RNAs is a prevalent source of genetic diversity amongst these isolates and is captured using Tiled-ClickSeq.

### Clinical isolates of SARS-CoV-2

We assayed nasopharyngeal (NP) swabs from 60 individuals who tested positive for SARS-CoV-2 in UTMB clinics between 5th January and 4th February 2021, which corresponds to the beginning of the 'third wave' driven by the B.1.1.7 '*Alpha*' variant of concern (VOC). CTs available for 51 of the samples ranged from 15 to 35 (*Source data 3*). RNA was extracted from 200 μl of the inactivated Viral Transport Medium (VTM) for each of these samples and used as input for Tiled-ClickSeq library synthesis using the v3 tiled primers and only 18 cycles in the PCR amplification step. Similar to the SARS-CoV-2 isolates, libraries were pooled and sequenced on either a MiSeq or NextSeq and raw data were processed using the computational pipeline established above.

The percentage coverage of the genome with greater than 10 reads is illustrated in *Figure 6A* as a function of the CT value (where available) for each sample and coloured according to the sequencing batch. At low CT values < 20, all but one sample returned complete genome coverage. Between CTs values of 20 and 25, coverage began to drop with the only one complete genome being obtained with a CT >30. The raw read coverage of the Tiled-ClickSeq data is shown for each sample in *Figure 6— figure supplement 1*. At low CTs ( < 17.5), genome coverage ranging from 100x to 10,000x is seen

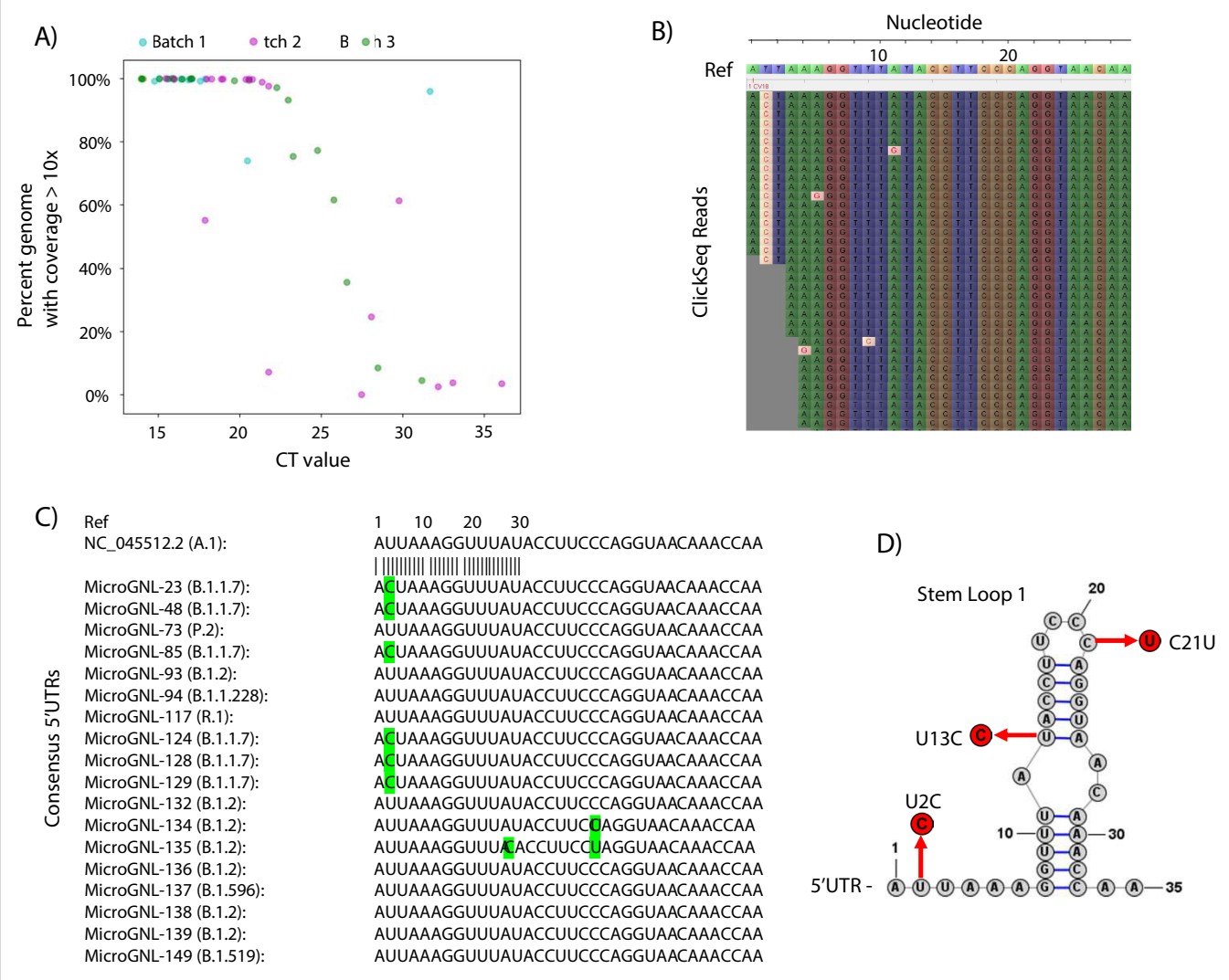

**Figure 6.** Tiled-ClickSeq for surveillance of SARS-CoV-2 from nasopharyngeal (NP) swabs collected during routine diagnostics of COVID19 at UTMB. (**A**) The percent genome coverage with greater than 10 reads is plotted as a function of the measured CT value (x-axis) for each clinical sample sequenced. Each point is colour-coded according to the batch of NGS libraries synthesized. (**B**) Sequence reads for one of the SARS-CoV-2 samples from the B.1.1.7 lineage are illustrated using Tablet sequence viewer to indicate the U to C transition at nt 2 (U2C) of the SARS-CoV-2 genome. (**C**) Sequence alignments of the 5'UTR of the consensus genomes of 18 clinical samples assayed illustrates the U2C SNVs found in each B.1.1.7 variant as well as a U13C and C21U in two other B.1.2 variants. (**D**) The structure of the first 35 nts of the SARS-CoV-2 5'UTR is illustrated which contains Stem Loop 1. The three SNVs identified in the consensus genomes of clinical samples are indicated.

The online version of this article includes the following figure supplement(s) for figure 6:

**Figure supplement 1.** Read coverage of Tiled-ClickSeq data over 60 SARS-CoV-2 clinical specimens: Read coverage obtained from Tiled-ClickSeq over the whole viral genome for 60 SARS-CoV-2 clinical specimens is depicted when sequenced on an Illumina NextSeq.

for all the samples for almost the entire genome. Coverage reduces corresponding with increased CT values, with only patchy coverage observed for samples with CT >30. Therefore, from a total of 60 samples, we obtained 36 complete genomes passing quality filters ( > 99.5% total genome coverage) which were deposited into GISAID. The majority of these were assigned to the B.1.2 *Pango* lineage (*Source data 3*).

As Tiled-ClickSeq can capture the entire genome including the 5'UTR, we inspected the raw data mapping to the reconstructed references to verify the SNV assignment. We found multiple isolates with SNVs in the 5'UTR. From the 36 reconstructed genomes, 8 were from the B.1.1.7 lineage (*Source data 3*). Although read coverage is generally diminished at the 5' ends, 7 of these had sufficient coverage ( > 10 x coverage) to assign a U-to-C transition at the second nucleotide of the genome

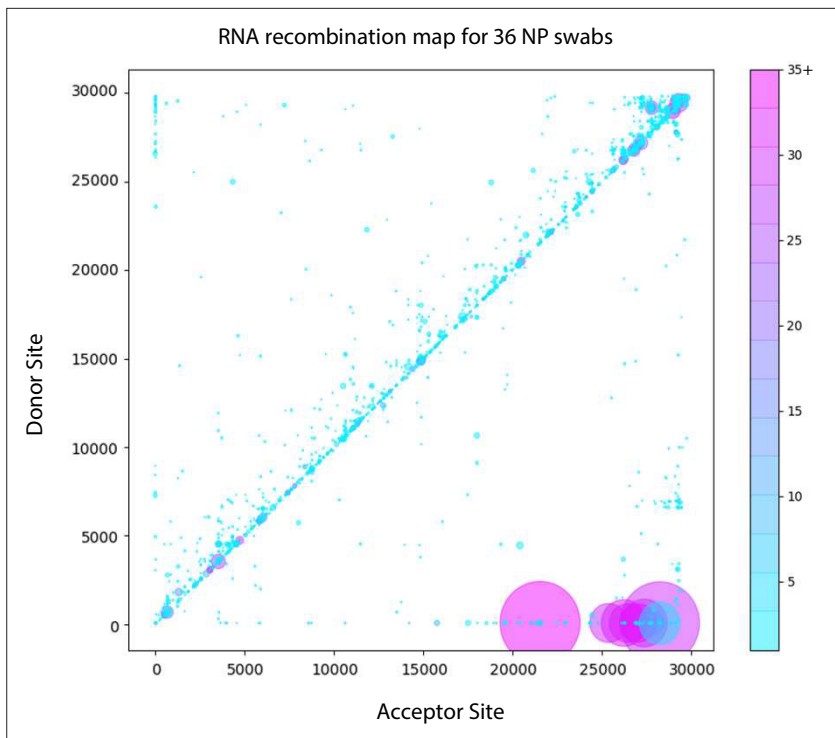

**Figure 7.** Tiled-ClickSeq identifies sub-genomic mRNAs, structural variants and Defective-RNAs in clinical samples of SARS-CoV-2. Similarly to *Figure 5B*, unique RNA recombination events are plotted for 36 clinical samples as a scatterplot whereby the upstream 'donor' site is plotted on the y-axis and a downstream 'acceptor' site is plotted on x-axis using the WA-1 reference coordinates for each sample. The read count for each unique RNA recombination event is indicated by the size of the point, while the number of samples in which this each RNA recombination event is found is indicated by the colour-bar. Insertions/duplication/back-splicing events are found above the x = y axis, while deletions and RNA recombination events yielding sgmRNAs are found below.

The online version of this article includes the following figure supplement(s) for figure 7:

**Source data 1.** BED files of RNA recombination events detected by ViReMa in the Tiled-ClickSeq data from each WRCEVA isolate and clinical sample.

**Figure supplement 1.** Tiled-ClickSeq identifies sub-genomic mRNAs, structural variants of clinical samples of SARS-CoV-2.

(U2C) (*Figure 6B and C*). To determine whether this was a novel SNV, we queried all SARS-CoV-2 genomes deposited in the GISAID database (2'867'767 genomes on the 19th August 2021) and extracted the first 30 nts of each genome. Only 176 genomes contained the U2C SNV; 70 of these were from B.1.1.7 '*Alpha*' VOC and 43 were from the B.1.617.2 '*Delta*' VOC. We additionally found C21U in MicroGNL_134 and U13C and C21U in MicroGNL_135. Both of these SNVs are located in the first stem-loop of the SNV 5'UTR (SL1) (*Figure 6D*).

Recombination analysis using *ViReMa* revealed all the expected sgmRNAs (*Figure 7* and *Figure 7—figure supplement 1*, *Figure 7—source data 1*). Similar to the distribution of RNA recombination frequencies observed for the cell-culture isolates (both here and in our previous studies [*Gribble et al., 2021*]), we observe an enrichment of small recombination events near the 3'UTR of SARS-CoV-2. The two common recombination events in this region that correspond to small duplications (29173^27,802 and 29442^29323, using WA-1 coordinates, indicated *Figure 7*) were found in 14 and 17 of the clinical samples respectively (*Figure 7—figure supplement 1*) at frequencies ranging up to approximately 8 % of the sequencing reads mapping in these regions. Interestingly, although it is a conserved feature in clinical isolates, it is not found in any of the cell-culture derived samples assayed (*Figure 4—source data 1*). Larger RNA recombination events corresponding to D-RNAs similar to those we have previously characterized in samples derived from cell-culture are also observed. These predominantly comprise recombination events with donor sites at nt 6000–7000 and acceptor sites in the 3'UTRs. Interestingly, only one sample (MicroGNL-163: B.1.2) had any evidence of InDels at

the furin cleavage site of Spike protein with 83 unique reads mapping over a 23583^23,599 deletion, corresponding to an approximate frequency of 5 % of the total virus reads mapping across this locus.

## Discussion

Tiled-ClickSeq provides a simple method for whole genome sequencing of virus isolates such as SARS-CoV-2 that can simultaneously map SNVs, minority variants as well as recombination events. Importantly, having only a single template-targeted primer per amplicon provides the opportunity to sequence any RNA template regardless of what expected or unknown sequence is found upstream, including recombinant RNA molecules such as sgmRNAs and D-RNAs. The targeted approach requires a relatively small number of reads to be collected, allowing 10 s of samples to be processed on a MiSeq platform or potentially 100 s on a single flowcell of a NextSeq. Furthermore, the same library preps can be used as input in Oxford Nanopore Sequencing pipelines to yield longer reads, providing the convenience and portability inherent to the platform. We demonstrated that this method can reconstruct full-length SARS-CoV-2 genomes in a manner equivalent to random-primed methods. Full length-genome sequencing is achieved, including the 5'UTR which is missed in the bulk of current high-throughput sequencing efforts, removing the need for 5'RACE.

We further validated our method by sequencing SARS-CoV-2 genomes from discarded clinical specimens (nasopharyngeal swabs in viral transport media) used for routine COVID19 diagnostics. From 60 specimens, we obtained 36 consensus genomes with greater than 10 x coverage across >99.5% of the viral genome. With only 18 PCR cycles, genome coverage began to drop concomitantly with the viral titer of the specimen, with samples with a CT value lower than 25 providing the best quality sequence data. From these samples, we found that all the B.1.1.7 variants characterized contained a U2C SNV. It is yet to be determined whether this SNV is unique to clinical samples collected in the geographical region of this study and the small number of SARS-CoV-2 deposited in GISAID that resolved the 5'UTRs. It is possible many B.1.1.7 isolates/samples bear this mutation but have been missed due to the prevalent use of paired-primer approaches whose 5'-most primer is downstream of or annealing to this locus. In either case, this may be significant as the 5'UTR contains many important regulatory elements that impact replication and translation as well as recombination events that generate the viral sgmRNAs (*Li et al., 2008*; *Zuniga et al., 2004*) and therefore may impact viral fitness.

An additional interesting feature of the clinical specimens was in the absence of small InDels in and adjacent to the furin cleavage site of Spike. Small deletions such as the actual furin cleave site 'PRRA' and the 'QTQTN' motif upstream of the site are prevalent among cell-passaged and amplified isolates of SARS-CoV-2 (*Liu et al., 2020*). Indeed, we recently demonstrated that these deletions confer a strong selection advantage in Vero E6 cells, rationalizing their common occurrence in in vitro preparations (*Johnson et al., 2021*). It is surprising therefore that despite observing these deletions in almost all of the cell-culture isolates to some small degree, we did not find any evidence of small deletions in the 60 clinical specimens analyzed here with the exception of one sample (MicroGNL_163). This suggests that while SARS-CoV-2 may be poised to generate structural variants in this region, they are not observed in primary human samples and thus not selected for during human infections.

Recombination analysis using *ViReMa* confirms that D-RNAs are a true component of the intrahost diversity of clinical specimens of SARS-CoV-2, albeit at low abundance and are highly variable from sample to sample in terms of their precise junctions. In contrast, we observed a small number of unique recombination events in multiple clinical samples, such as 29442^29,323. The 29442^29,323 event is a 120nt duplication that maintains the open reading frame in the C-terminal end of Nucleocapsid (N) at residues 350–389, which contains the last 10 structured amino acids of the C-terminal domain (*Peng et al., 2020*). Interestingly, this event is not found in any of the cell-culture derived samples suggesting that it is purged during expansion and isolation. Multiple small insertions and duplications have been previously reported in this locus of SARS-CoV-2 (*Garushyants et al., 2021*), although these reported events are smaller than those detected here ( ≤ 24 nts). Genomic duplication followed by 'resolution' through a series of recombination events is a common mechanism observed in RNA viruses that allows for virus diversification and evolution of the duplicated regions (*Scheel et al., 2013*; *Bentley et al., 2021*). Such large duplications may not be resolved when using paired-primer genomics approaches as these would generate amplicons larger than expected and so may be filtered out during final NGS library size selection or during bioinformatic processing.

The design of Tiled-ClickSeq imparts built-in quality control tools, including UMIs in the click-adaptor and the opportunity to use paired-end sequencing to identify the primer that gives rise to each amplicon. In addition to controlling for aberrant SNVs, minority and structural variants, this information can be used to determine the relative sensitivity and specificity of each primer in the primer mix allowing the scheme to be pruned and optimized. Our final primer scheme contained over 390 unique SARS-CoV-2 primers. This purposely thorough design demonstrates how the Tiled-ClickSeq pipeline can accommodate complex mixtures of overlapping primers within the same RT reaction. This built-in redundancy reduces the chance of primer dropout due to the presence of SNVs, SVs or recombination events found in primer-annealing sites. This feature is especially important considering the emergence of SARS-CoV-2 variants with deletions and mutations that disrupt sequencing efforts (*Plante et al., 2021*). Interestingly, we detected very few minority variants in our samples present above 2 %. This is consistent with other reports of minority variant detection in SARS-CoV-2 isolates and likely reflects the well-characterized activity of the coronavirus ExoN-nsp14 as a 'proof-reader' enzyme (*Smith and Denison, 2013*). As a result, the greatest source of genetic diversity in coronavirus isolates may well be due to RNA recombination.

On the nanopore sequencing platform, we could obtain sequence reads within the same day as RNA extraction. While the baseline accuracy rate of the nanopore platform prevents the reliable annotation of minority variants present at <5%, this platform can reconstruct novel SARS-CoV-2 variants as well as identify abundant sgmRNAs. Nanopore sequencing also allows for identification of long-range epistatically linked variants (*Gallardo et al., 2021*). Epistatic linkage can also be computationally leveraged to identify minority variants present at levels below the baseline error-rate of the sequencing platform, for example, using *CliqueSNV* (*Knyazev et al., 2020*) or *CoVaMa* (*Routh et al., 2015a*). Therefore, the nanopore platform in combination with Tiled-ClickSeq provides a robust pipeline for high-throughput SARS-CoV-2 variant detection with minimal infrastructure.

The ARTIC protocol contains a primer cognate to the 5'UTR of SARS-CoV-2 (nts 30–54) to capture and quantitate sub-genomic mRNAs (*Parker et al., 2020*). However, recombination events including non-canonical sgmRNAs will be missed by primer-pools that do not happen to flank RNA recombination junctions. In contrast, Tiled-ClickSeq is capable of 'agnostically' detecting any unanticipated RNA recombination including D-RNAs that can be characterized by RNA recombination events in or between any region of the viral genome, often in an unpredictable manner. As ClickSeq was originally designed to avoid artefactual recombination with fewer than three artefactual chimeric reads found per million reads, Tiled-ClickSeq provides a useful tool to identify D-RNAs and to robustly characterize rates of RNA recombination. Together, using the Tiled ClickSeq approach, we have the opportunity to identify rare and unexpected recombination events and are not biased by the limitation of primer-pair approaches. Coupled with its cross-sequencing platform capabilities, the work highlights the utility of Tiled-ClickSeq for analysis of SARS-CoV-2.

## Materials and methods

**Key resources table**

| Reagent type (species) or resource | Designation | Source or reference | Identifiers | Additional information |
|---|---|---|---|---|
| Chemical compound, drug | AzNTPs | BaseClick | • BCT-25, BCT-26, BCT-27, BCT-28 | |
| Sequence-based reagent | 5'-hexynyl-functionalized i5 oligo | IDT | • DNA oligo | |
| Sequence-based reagent | TCSv3 | This Paper | PCR Primers | *Source data 1* |
| Software, algorithm | Bash and python3 text files | This Paper | .txt and.py files | *Source data 2* |

### Viruses and RNA extraction

For the World Reference Center for Emerging Viruses and Arboviruses (WRCEVA) isolates, viral RNA was obtained from supernatant materials of viral isolates amplified on Vero cells originally obtained from nasopharyngeal swab samples that tested positive in clinical laboratory assays for SARS-CoV-2 RNA, as described previously (*Harcourt et al., 2020b*). The use of deidentified human samples was approved by the UTMB IRB under protocol 20–0088. The recombinant wild-type and 'PRRA-deletion'

mutant SARS-CoV-2 are based on the sequence of USA-WA1/2020 isolate provided by the WRCEVA as previously described (*Xie et al., 2020*; *Johnson et al., 2021*). Wild-type and mutant SARS-CoV-2 were titrated and propagated on Vero E6 cells. RNAs were extracted from either total cellular materials or supernatants as indicated in the main text.

## Collection of RNA from clinical samples

Discarded nasopharyngeal swabs Viral Transport Media (VTM) used for COVID19 diagnostics at UTMB were collected and de-identified before being discarded. Only the CT value was retained. Of the remaining VTM, 200 µl was put into 1000 µl of Trizol LS for inactivation and stored at –80 °C overnight. The next day, samples were thawed, vortexed and 266.6 µl of chloroform was added. Samples were centrifuged at 12,500 g for 15 min at 4 °C to separate the phases. The top aqueous phase (400 µl-450µl) was aspirated and added to 667 µl isopropanol containing 2 µl of *glycoblue* and vortexed for 10 s. Next, samples were incubated at room temperature for 10 minutes and centrifuged at 20,800 g for 20 min at 4 °C to pellet RNA. Isopropanol was removed and 1 ml 75 % ethanol was added, vortexed and inverted several times. RNA was precipitated by centrifugation at 10,000 g for 10 min at 4 °C. Ethanol was removed and the pellet allowed to air-dry for 5–10 min. The RNA pellet was then resuspended in 20 µl of RNase/DNase free water. Five µl of the resuspended RNA was subsequently used as input in Tiled-ClickSeq library preparation.

## SARS-CoV-2 reverse transcription primer design

A 'first' tiled-primer set (v1) containing 71 primers was designed cognate to the WA-1 SARS-CoV-2 genome (accession number: NC_045512.2) using the *primalseq* webserver (*Grubaugh et al., 2019a*; http://primal.zibraproject.org/) with an amplicon distance of approximately 500nt in between each primer pair. We used only the 'right' primer sequences generated by *primalseq* and appended the Illumina p7 adaptor to these (e.g. GTGACTGGAGTTCAGACGTGTGCTCTTCCGATCT+ NNNN + TGTC TCACCACTACGACCGTAC). We also included an additional primer designed to target the 3'-most 25 nts of the SARS-CoV-2 genome. A 'second' tiled-primer set was synthesized in a similar fashion using 326 loci described previously (*Guo et al., 2020*). A 'third' tiled-primer pool (v3) was generated by combining the v1 and v2 pools. Primers used in this study are provided as BED files with their loci and the corresponding sequence in *Source data 1*. Each primer was pooled in equimolar ratios to yield a SARS-CoV-2 specific primer pool used for the RT step of Tiled-ClickSeq.

## ClickSeq library preps

Random-primed ClickSeq NGS libraries were synthesized as described in previously published protocols from our lab (*Jaworski and Routh, 2018*; *Routh et al., 2015b*). In *Jaworski and Routh, 2018*, MiMB (*Jaworski and Routh, 2018*), we provided detailed descriptions of each step of the protocol and emphasize issues that affect the success and quality of the final libraries. Here, for Tiled-ClickSeq, the protocol has two important adjustments: (1) firstly, the primers used to initiate reverse transcription comprised pools of 10 s-100s of virus-specific primer oligos (primer pool v1, v2, and/or v3, described above); and (2) we annealed the RT-primers to the RNA template by incubating the RNA and primer mixture at 65 °C for 5 min, followed by a slow-cool of 1 degree per second to a final temperature or 12 °C. RT-enzyme mixes were added at 12 °C and primer extension is performed for 10 min at 55 °C. All subsequent steps of the Tiled-ClickSeq reaction, comprising RT cleanup, click-ligation, PCR amplification and cDNA library size-selection are identical to those used in the random-primed ClickSeq method and described previously (*Jaworski and Routh, 2018*). The i5 'Click-Adaptor' was a reverse complement of the full Illumina Universal Adaptor sequence, plus an additional twelve 'N's at 5'-end to provide a Unique Molecular Identifier and functionalized with a 5'-hexynyl group (Integrated DNA Technologies). Final NGS libraries containing fragment sizes ranging 300–700 nts were pooled and sequenced on Illumina MiSeq, MiniSeq or NextSeq platforms using paired-end sequencing.

## ARTIC amplicon sequencing

We used the standard protocol for ARTIC sequencing of SARS-CoV-2 as is well described (*Tyson et al., 2020*; dx.doi.org/10.17504/protocols.io.bbmuik6w, last accessed 26th Aug 2021) using the version3 ARTIC primer sets. Briefly, cDNA was generated using random-primed reverse-transcription reactions. DNA amplicons 300-500nts in length were generated in two separate PCR reactions containing pairs

of primers targeting the SARS-CoV-2 that together provide overlapping amplicons to generate full-length genome sequence coverage. DNA amplicons were pooled in equimolar ratios and we used the Illumina NEBNext Multiplexed sequencing kit to append barcoded adaptors and performed 2 × 150 nt PE sequencing. Whole SARS-CoV-2 genomes were reconstructed by mapping both forward and reverse reads to the reference WA-1 strain and calling SNVs using *pilon, RRID:SCR_014731* (*Walker et al., 2014*), in an identical fashion performed for the Tiled-ClickSeq data.

## Nanopore sequencing

Final cDNA libraries generated by the Tiled-ClickSeq protocol, although containing Illumina adaptors, are compatible with the Direct Sequencing by Ligation Kit (LSK-109) provided by Oxford Nanopore Technologies. cDNA library fragments > 600 nts in length were gel extracted and processed for nanopore sequencing using the manufacturer's protocols. The addition of demultiplexing barcodes can be achieved using the Native Barcoding by Ligation module (NBD104), again following the manufacturer's protocols. Single-plex or pooled cDNA libraries with ONT adaptors were loaded onto MIN-FLO109 flowcells on a MinION Mk1C and sequenced using the MinKNOW controller software for >24 hours. Raw FAST5 reads were base-called and demultiplexed using *Guppy*.

## Bioinformatics

All batch scripts and custom python scripts used in this manuscript are available in *Source data 2*. Specific command-line entries and parameters can be found therein.

For Illumina reads, raw data were filtered and trimmed using *fastp, RRID:SCR_016962* (*Chen et al., 2018Chen et al., 2018*) to remove Illumina adaptors, quality filter reads and extract Unique Molecular Identifiers (UMIs). A custom python3 script was written to split the raw 'forward'/R1 reads into multiple individual FASTQ files depending upon the tiled-sequencing primer that is present in the first 30 nts of the 'reverse'/R2 paired-read (*Source data 2*). These split FASTQ files were then trimmed using *cutadapt, RRID:SCR_011841* (*Martin, 2011*) to remove primer-derived sequences from the R1 reads. After trimming, all the split R1 files were re-combined to yield a final processed dataset. These reads were mapped to the WA-1 strain (NC_045512.2) of SARS-CoV-2 using *bowtie2, RRID:SCR_016368* (*Langmead and Salzberg, 2012*) and a new reference consensus genome was rebuilt for each dataset using *pilon, RRID:SCR_014731* (*Walker et al., 2014*). Next, we mapped the processed read data to the reconstructed reference genome using *ViReMa, RRID:SCR_000566* (*Routh and Johnson, 2014*) to map to both the virus and the host (*chlSab2* or *hg19*) genome. SAM files were manipulated using *samtools, RRID:SCR_002105* (*Li et al., 2009*) and de-duplicated using *umi-tools, RRID:SCR_017048* (*Smith et al., 2017*). Minority variants were extracted using the *mpileup* command in *samtools, RRID:SCR_002105* (*Li et al., 2009*) and a custom python3 script to count nucleotide frequency at each coordinate to find minority variants (*Source data 2*). Mapped data were visualized using *Tablet, RRID:SCR_000017* (*Milne et al., 2010*).

For Nanopore reads, *porechop, RRID:SCR_016967* (https://github.com/rrwick/Porechop) was used to remove Illumina adaptor sequences and reads greater than 100nts in length were retained. These were mapped to the WA-1 SARS-CoV-2 genome (NC_045512.2) using *minimap2, RRID:SCR_018550* (*Li, 2016*) with the *-splice* option selected. Output SAM files were processed using *samtools, RRID:SCR_002105* (*Li et al., 2009Li et al., 2009*) and *bedtools, RRID:SCR_006646* (*Quinlan, 2014Quinlan, 2014*) to generate coverage maps.

## Acknowledgements

This work was funded by: a Technology Commercialization Program grant (UTMB) to EJ and ALR; a 'Data collection grant' and 'COVID-19 funding' from the Institute for Human Infections & Immunity (UTMB) to ALR; NIH grant R21AI151725 from to ALR; CDC contract 200-2021-11195 to ALR and S.W.; NIH Grant R01AI153602 and R00AG049042 to VDM; NIH grant R24 AI120942 "World Reference Center for Emerging Viruses and Arboviruses" to SW; and The Sealy and Smith Foundation for financial support to SW and UTMB. RML was supported by an NRSA Clinical and Translational Science (NCATS) TL-1 Training Core Award (TL1TR001440). We thank Brittany Dennler for graphic design support and figure illustrations.

# Additional information

## Competing interests

Elizabeth Jaworski: E.J. and A.R. are co-founders and owners of 'ClickSeq Technologies LLC', a Texas-based Next-Generation Sequencing provider offering ClickSeq kits and services including the methods described in this manuscript. E.J. and A.R have filed a patent application (PCT/US2021/038048) on the method and use of single-primer tiled sequencing.. The other authors declare that no competing interests exist.

## Funding

| Funder | Grant reference number | Author |
| --- | --- | --- |
| National Institutes of Health | R21AI151725 | Andrew L Routh |
| National Institutes of Health | R01AI153602 | Vineet D Menachery |
| National Institutes of Health | R00AG049042 | Vineet D Menachery |
| National Institutes of Health | R24AI120942 | Scott C Weaver |
| The Sealy and Smith Foundation | | Scott C Weaver |
| University of Texas Medical Branch at Galveston | | Andrew L Routh |
| Centers for Disease Control and Prevention | 200-2021-11195 | Scott Weaver Andrew L Routh |
| Institute for Translational Sciences, University of Texas Medical Branch | TL1TR001440 | Rose M Langsjoen |
| Fundação de Amparo à Pesquisa do Estado de São Paulo | Grant 2019/27803-2 | Rafael RG Machado |

The funders had no role in study design, data collection and interpretation, or the decision to submit the work for publication.

## Author contributions

Elizabeth Jaworski, Conceptualization, Data curation, Investigation, Methodology, Resources, Validation, Writing – review and editing; Rose M Langsjoen, Investigation, Methodology, Resources, Writing – review and editing; Brooke Mitchell, Barbara Judy, Patrick Newman, Jessica A Plante, Kenneth S Plante, Aaron L Miller, Stephanea Sotcheff, Victoria Morris, Jianli Dong, Ping Ren, Investigation, Resources; Yiyang Zhou, Daniele Swetnam, Data curation, Formal analysis, Validation, Visualization; Nehad Saada, Rafael RG Machado, Allan McConnell, Steven G Widen, Jill Thompson, Resources; Rick B Pyles, Thomas G Ksiazek, Resources, Supervision; Vineet D Menachery, Funding acquisition, Investigation, Resources, Supervision, Writing – review and editing; Scott C Weaver, Conceptualization, Data curation, Formal analysis, Funding acquisition, Investigation, Methodology, Project administration, Supervision, Writing – review and editing; Andrew L Routh, Conceptualization, Data curation, Formal analysis, Funding acquisition, Investigation, Methodology, Project administration, Software, Supervision, Validation, Visualization, Writing – original draft, Writing – review and editing

## Author ORCIDs

Jessica A Plante  http://orcid.org/0000-0002-4768-7458
Ping Ren  http://orcid.org/0000-0002-4022-6667
Andrew L Routh  http://orcid.org/0000-0002-2874-5990

## Decision letter and Author response

Decision letter https://doi.org/10.7554/eLife.68479.sa1

Author response https://doi.org/10.7554/eLife.68479.sa2

## Additional files

### Supplementary files

• Transparent reporting form

• Source data 1. Annotations and sequences of tiled-primers used in this manuscript are provided in BED format.

• Source data 2. Batch scripts provided all computational tools and parameters used and python3 scripts used in this study are provided.

• Source data 3. A summary of all Single-Nucleotide Variants (SNVs) detected for all samples sequenced in this study are provided. Each unique sample/isolate is listed, together with the SNVs relative to the WA-1 (NC_045512.2) strain in different NGS library preparation methods and sequencing platforms. The accession number for each reconstructed genome deposited in GenBank is also indicated.

### Data availability

All raw sequencing data (Illumina and Nanopore in FASTQ format) are available in the NCBI Small Read Archive with BioProject PRJNA707211. Consensus genomes for WRCEVA SARS-CoV-2 isolates reported in this manuscript are deposited at GenBank (MW047307-MW047318 and MW703487-MW703490). Genome sequences of clinical samples of SARS-CoV-2 were deposited in GISAID with accession numbers listed in SData 3 and are searchable in GISAID using 'clickseq'.

The following dataset was generated:

| Author(s) | Year | Dataset title | Dataset URL | Database and Identifier |
|---|---|---|---|---|
| Routh AL | 2020 | Tiled-ClickSeq Raw data | https://www.ncbi.nlm. nih.gov/sra/?term= PRJNA707211 | NCBI Sequence Read Archive, PRJNA707211 |
| Jaworski E, Langsjoen RM, Judy B, Newman P, Plante K, Pyles R, Ksiazek T, Weaver SC, Routh AL | 2021 | Severe acute respiratory syndrome coronavirus 2 isolate SARS-CoV-2/human/USA/ WRCEVA_000501/2020 | https://www.ncbi. nlm.nih.gov/nuccore/ MW047307 | NCBI Nucleotide, MW047307 |
| Jaworski E, Langsjoen RM, Judy B, Newman P, Plante K, Pyles R, Ksiazek T, Weaver SC, Routh AL | 2021 | Severe acute respiratory syndrome coronavirus 2 isolate SARS-CoV-2/human/USA/ WRCEVA_000502/2020 | https://www.ncbi. nlm.nih.gov/nuccore/ MW047308 | NCBI Nucleotide, MW047308 |
| Jaworski E, Langsjoen RM, Judy B, Newman P, Plante K, Pyles R, Ksiazek T, Weaver SC, Routh AL | 2021 | Severe acute respiratory syndrome coronavirus 2 isolate SARS-CoV-2/human/USA/ WRCEVA_000505/2020 | https://www.ncbi. nlm.nih.gov/nuccore/ MW047309 | NCBI Nucleotide, MW047309 |
| Jaworski E, Langsjoen RM, Judy B, Newman P, Plante K, Pyles R, Ksiazek T, Weaver SC, Routh AL | 2021 | Severe acute respiratory syndrome coronavirus 2 isolate SARS-CoV-2/human/USA/ WRCEVA_000506/2020 | https://www.ncbi. nlm.nih.gov/nuccore/ MW047310 | NCBI Nucleotide, MW047310 |
| Jaworski E, Langsjoen RM, Judy B, Newman P, Plante K, Pyles R, Ksiazek T, Weaver SC, Routh AL | 2021 | Severe acute respiratory syndrome coronavirus 2 isolate SARS-CoV-2/human/USA/ WRCEVA_000507/2020 | https://www.ncbi. nlm.nih.gov/nuccore/ MW047311 | NCBI Nucleotide, MW047311 |

*Continued on next page*

*Continued*

| Author(s) | Year | Dataset title | Dataset URL | Database and Identifier |
|---|---|---|---|---|
| Jaworski E, Langsjoen RM, Judy B, Newman P, Plante K, Pyles R, Ksiazek T, Weaver SC, Routh AL | 2021 | Severe acute respiratory syndrome coronavirus 2 isolate SARS-CoV-2/human/USA/WRCEVA_000508/2020 | https://www.ncbi.nlm.nih.gov/nuccore/MW047312 | NCBI Nucleotide, MW047312 |
| Jaworski E, Langsjoen RM, Judy B, Newman P, Plante K, Pyles R, Ksiazek T, Weaver SC, Routh AL | 2021 | Severe acute respiratory syndrome coronavirus 2 isolate SARS-CoV-2/human/USA/WRCEVA_000509/2020 | https://www.ncbi.nlm.nih.gov/nuccore/MW047313 | NCBI Nucleotide, MW047313 |
| Jaworski E, Langsjoen RM, Judy B, Newman P, Plante K, Pyles R, Ksiazek T, Weaver SC, Routh AL | 2021 | Severe acute respiratory syndrome coronavirus 2 isolate SARS-CoV-2/human/USA/WRCEVA_000510/2020 | https://www.ncbi.nlm.nih.gov/nuccore/MW047314 | NCBI Nucleotide, MW047314 |
| Jaworski E, Langsjoen RM, Judy B, Newman P, Plante K, Pyles R, Ksiazek T, Weaver SC, Routh AL | 2021 | Severe acute respiratory syndrome coronavirus 2 isolate SARS-CoV-2/human/USA/WRCEVA_000513/2020 | https://www.ncbi.nlm.nih.gov/nuccore/MW047315 | NCBI Nucleotide, MW047315 |
| Jaworski E, Langsjoen RM, Judy B, Newman P, Plante K, Pyles R, Ksiazek T, Weaver SC, Routh AL | 2021 | Severe acute respiratory syndrome coronavirus 2 isolate SARS-CoV-2/human/USA/WRCEVA_000514/2020 | https://www.ncbi.nlm.nih.gov/nuccore/MW047316 | NCBI Nucleotide, MW047316 |
| Jaworski E, Langsjoen RM, Judy B, Newman P, Plante K, Pyles R, Ksiazek T, Weaver SC, Routh AL | 2021 | Severe acute respiratory syndrome coronavirus 2 isolate SARS-CoV-2/human/USA/WRCEVA_000515/2020 | https://www.ncbi.nlm.nih.gov/nuccore/MW047317 | NCBI Nucleotide, MW047317 |
| Jaworski E, Langsjoen RM, Judy B, Newman P, Plante K, Pyles R, Ksiazek T, Weaver SC, Routh AL | 2021 | Severe acute respiratory syndrome coronavirus 2 isolate SARS-CoV-2/human/USA/WRCEVA_000516/2020 | https://www.ncbi.nlm.nih.gov/nuccore/MW047318 | NCBI Nucleotide, MW047318 |
| Jaworski E, Langsjoen RM, Judy B, Newman P, Plante K, Pyles R, Ksiazek T, Weaver SC, Routh AL | 2021 | Severe acute respiratory syndrome coronavirus 2 isolate SARS-CoV-2/human/USA/WRCEVA_000503/2020 | https://www.ncbi.nlm.nih.gov/nuccore/MW703487 | NCBI Nucleotide, MW703487 |
| Jaworski E, Langsjoen RM, Judy B, Newman P, Plante K, Pyles R, Ksiazek T, Weaver SC, Routh AL | 2021 | Severe acute respiratory syndrome coronavirus 2 isolate SARS-CoV-2/human/USA/WRCEVA_000504/2020 | https://www.ncbi.nlm.nih.gov/nuccore/MW703488 | NCBI Nucleotide, MW703488 |
| Jaworski E, Langsjoen RM, Judy B, Newman P, Plante K, Pyles R, Ksiazek T, Weaver SC, Routh AL | 2021 | Severe acute respiratory syndrome coronavirus 2 isolate SARS-CoV-2/human/USA/WRCEVA_000511/2020 | https://www.ncbi.nlm.nih.gov/nuccore/MW703489 | NCBI Nucleotide, MW703489 |
| Jaworski E, Langsjoen RM, Judy B, Newman P, Plante K, Pyles R, Ksiazek T, Weaver SC, Routh AL | 2021 | Severe acute respiratory syndrome coronavirus 2 isolate SARS-CoV-2/human/USA/WRCEVA_000512/2020 | https://www.ncbi.nlm.nih.gov/nuccore/MW703490 | NCBI Nucleotide, MW703490 |

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
