## [Decision Letter]

**Acceptance summary:**

The proposed method provides an alternate approach for viral whole genome sequencing that appears to have advantages over current amplicon methods in terms of detecting variants such as duplications. This may be important to explore viral variants and evolution.

**Decision letter after peer review:**

Thank you for submitting your article "Tiled-ClickSeq for targeted sequencing of complete coronavirus genomes with simultaneous capture of RNA recombination and minority variants" for consideration by *eLife*. Your article has been reviewed by 3 peer reviewers, including Anurag Agrawal as Reviewing editor and Reviewer #1, and the evaluation has been overseen by K VijayRaghavan as the Senior Editor.

While the work is of interest and the proposed new method seems useful, extensive additional data is required, specifically by comparisons to existing approaches in clinical samples, since the value of this approach can only be judged after seeing the additional data.

Essential revisions:

1) The method must be tested on actual clinical samples, with comparison to at least one of the existing methods in wide use, so that claims of better performance can be judged with evidence. Average read depth (relative to total nucleotides sequenced per sample) across multiple samples with confidence intervals or equivalent to visualize run-to-run variability should be shown in comparison to other approaches.

2) Provide evidence to support the claim of identifying recombinant RNAs that are otherwise undetectable.

*Reviewer #1 (Recommendations for the authors):*

1. The claims for sensitivity and specificity must be backed by actual comparisons.

2. Is there an example of a recombinant RNA identified by this approach missed by alternative approaches?

*Reviewer #2 (Recommendations for the authors):*

I would be much more supportive of this manuscript if the authors were able to expand on the current body of knowledge about SARS-CoV-2 using their sequencing method, even if this is limited to identifying recombinant RNAs that are otherwise undetectable.

*Reviewer #3 (Recommendations for the authors):*

The main critique and weakness are that the authors did not use the method described to sequence directly from the primary nasopharyngeal samples, but rather from Vero cell-grown SARS-CoV-2 isolates. This has dampened the significance of this potentially important contribution and should be easily addressed in a revised form. One of the greatest obstacles in the viral sequencing space, especially with the emergence of SARS-CoV-2 variants, is in resolving the sequence of primary samples and not from cell grown-cells. A lot of public laboratories may not have the time, resource or expertise to grow viral isolates in Vero cells first and then sequence. If the authors can provide evidence that their approach is comparable or perhaps better than the current method (i.e., ARTIC protocol) in generating sequences directly from samples then this method can significantly contribute to field.

---

## [Author Response]

Essential revisions:1) The method must be tested on actual clinical samples, with comparison to at least one of the existing methods in wide use, so that claims of better performance can be judged with evidence. Average read depth (relative to total nucleotides sequenced per sample) across multiple samples with confidence intervals or equivalent to visualize run-to-run variability should be shown in comparison to other approaches

We have provided a substantial update to the manuscript by providing an analysis of 60 clinical specimens of SARS-CoV-2, as described below in the detailed responses to the reviewers that addresses all of these issues.

2) Provide evidence to support the claim of identifying recombinant RNAs that are otherwise undetectable.

In the revision, we describe the recombination analysis of multiple clinical samples of SARS-CoV-2. To illustrate the potential of Tiled-ClickSeq to characterize novel recombination events, we describe the observation of a large genome duplication (annotated as 29442^29323) found in multiple clinical samples, but not any cell-culture samples (providing support that these are not sequence artifacts). We are not so bold as to state that ARTIC approaches or other could never detect such an event, but simply state that this event is an example of one that has not been observed before.

In the manuscript, we now present data using the ARTIC protocol and Illumina sequencing to study the 12 WRCEVA isolates described in the manuscript and presented in Figure 3. We performed an analysis of recombination events using a similar pipeline to that described for the Tiled-ClickSeq data. After adaptor trimming and quality filtering the read data, recombination events were characterized using ViReMa. The recombination heatmaps (analogous to Figure 5B) for these data are shown in Author response image 1.

**Author response image 1. sa2fig1:** Similarly to Figure 5B in the manuscript, unique RNA recombination events are plotted for WRCEVA samples as a scatter plots whereby the upstream ‘donor’ site is plotted on the y-axis and a downstream ‘acceptor’ site is plotted on x-axis. The read count for each unique RNA recombination event is indicated by the size of the point, while the number of samples in which this each RNA recombination event is found is indicated by the colour-bar. Insertions/duplication/back-splicing events are found above the x=y axis, while deletions and RNA recombination events yielding sgmRNAs are found in Author response image 2.

Similarly to Figure 5B in the manuscript, unique RNA recombination events are plotted for WRCEVA samples as a scatter plot whereby the upstream ‘donor’ site is plotted on the y-axis and a downstream ‘acceptor’ site is plotted on x-axis. The read count for each unique RNA recombination event is indicated by the size of the point, while the number of samples in which this each RNA recombination event is found is indicated by the colour-bar.Insertions/duplication/back-splicing events are found above the x=y axis, while deletions and RNA recombination events yielding sgmRNAs are found in Author response image 2.

While this does indeed capture the expected sgmRNAs (large purple circles in bottom right), there are many spurious recombination events also reported. These are represented by the characteristic vertical and horizontal striations present in this heatmap and which are absent in our Tiled-ClickSeq data. As we suspected that these events are derived from mis-priming events in during the ARTIC PCR step, we additionally trimmed the first 25 nts from each read and re-mapped the data using ViReMa. A new heat map is shown in Author response image 2.

**Author response image 2. sa2fig2:** Same as Author response image 1, except that read data used in the analysis have been trimmed by 25nts to remove any potential primer-derived sequences that may give rise to artifactual chimeric reads.

This removes some of the striations, indicating that recombination/chimeric artefacts have been removed. However, many events remain, in particular we still see numerous small InDels that are not found in the Tiled-ClickSeq data. Furthermore, this primer-trimming results in the loss of the detection of the sgmRNAs. This is likely due to the closeness of the primer annealing site (nts30-54) to the sgmRNAs recombination site (nt70). Trimming these 24 nts from the reads will leave only ~14nts of the Leader sequence, which is shorter than the seed length used in ViReMa.

These issues do not demonstrate that ARTIC or similar approaches cannot be used for recombination analysis with careful optimization or primer designs, but merely illustrate that there is substantial background noise and that analyses are at least problematic. We feel that it would be infeasible to exhaustively demonstrate that other approaches cannot solve recombination events, and indeed the capture of these may vary greatly among different experimentalists using different wet-lab and bioinformatics protocols.

Rather, we hope that in the revised manuscript, the power of Tiled-ClickSeq to reliably detect and report novel recombination events is clear as presented. We also emphasize in the manuscript how Tiled-ClickSeq has the capacity to resolve the entire 5’ UTR of SARS-CoV-2 genomes, which paired-primer amplicons inherently cannot. We found a U2C SNV that is pervasive throughout all B.1.1.7 variants sequenced here. This is an interestingly and novel feature that we are following up on. Altogether, we believe that Tiled-ClickSeq provides a unique and flexible platform for targeted virus genomics that will have applications in many settings beyond those presented here.

Reviewer #1 (Recommendations for the authors):1. The claims for sensitivity and specificity must be backed by actual comparisons.

As described above, we provide a comprehensive analysis of clinical specimens and report the sensitivity and specificity of our approach. These new data are presented in Figure 6 and described in the main text.

2. Is there an example of a recombinant RNA identified by this approach missed by alternative approaches?

In the manuscript we provide an analysis of RNA recombination events found in clinical specimens of SARS-CoV-2 wherein we observe a pervasive 29422^29323 duplication. We do not feel we can state that this event cannot be captured using other approaches (indeed it would not be feasible to exhaustively demonstrate this), however, we simply state that it has not been previously observed despite being so prevalent here. These new data we presented in Figure 7 and supplementary materials and discussed in the main text.

Reviewer #2 (Recommendations for the authors):I would be much more supportive of this manuscript if the authors were able to expand on the current body of knowledge about SARS-CoV-2 using their sequencing method, even if this is limited to identifying recombinant RNAs that are otherwise undetectable.

Thank you for your comments. In the revised manuscript we have provided substantial additional information as described above with an analysis of multiple clinical samples of SARS-CoV-2. In particular, we identify SNVs in 5’UTRs and novel recombinant RNAs that have not previously been described.

Reviewer #3 (Recommendations for the authors):The main critique and weakness are that the authors did not use the method described to sequence directly from the primary nasopharyngeal samples, but rather from Vero cell-grown SARS-CoV-2 isolates. This has dampened the significance of this potentially important contribution and should be easily addressed in a revised form. One of the greatest obstacles in the viral sequencing space, especially with the emergence of SARS-CoV-2 variants, is in resolving the sequence of primary samples and not from cell grown-cells. A lot of public laboratories may not have the time, resource or expertise to grow viral isolates in Vero cells first and then sequence. If the authors can provide evidence that their approach is comparable or perhaps better than the current method (i.e., ARTIC protocol) in generating sequences directly from samples then this method can significantly contribute to field.

Thank you for your comments. We have addressed this issue in the revised manuscript.